# Evaluation of Current Amikacin Dosing Recommendations and Development of an Interactive Nomogram: The Role of Albumin

**DOI:** 10.3390/pharmaceutics13020264

**Published:** 2021-02-15

**Authors:** Jonás Samuel Pérez-Blanco, Eva María Sáez Fernández, María Victoria Calvo, José M. Lanao, Ana Martín-Suárez

**Affiliations:** 1Department of Pharmaceutical Sciences, Pharmacy Faculty, University of Salamanca, 37007 Salamanca, Spain; jsperez@usal.es (J.S.P.-B.); emsaez@saludcastillayleon.es (E.M.S.F.); toyi@usal.es (M.V.C.); amasu@usal.es (A.M.-S.); 2Institute for Biomedical Research of Salamanca (IBSAL), Paseo de San Vicente, 58-182, 37007 Salamanca, Spain; 3Pharmacy Service, University Hospital of Salamanca, Paseo de San Vicente, 58-182, 37007 Salamanca, Spain

**Keywords:** amikacin dosing recommendations guidelines, albumin, nomogram, model-informed precision dosing, pharmacokinetics/pharmacodynamics

## Abstract

This study aimed to evaluate the potential efficacy and safety of the amikacin dosage proposed by the main guidelines and to develop an interactive nomogram, especially focused on the potential impact of albumin on initial dosage recommendation. The probability of target attainment (PTA) for each of the different dosing recommendations was calculated through stochastic simulations based on pharmacokinetic/pharmacodynamic (PKPD) criteria. Large efficacy and safety differences were observed for the evaluated amikacin dosing guidelines together with a significant impact of albumin concentrations on efficacy and safety. For all recommended dosages evaluated, efficacy and safety criteria of amikacin dosage proposed were not achieved simultaneously in most of the clinical scenarios evaluated. Furthermore, a significant impact of albumin was identified: The higher is the albumin, (i) the higher will be the PTA for maximum concentration/minimum inhibitory concentration (Cmax/MIC), (ii) the lower will be the PTA for the time period with drug concentration exceeding MIC (T_>MIC_) and (iii) the lower will be the PTA for toxicity (minimum concentration). Thus, accounting for albumin effect might be of interest for future amikacin dosing guidelines updates. In addition, AMKnom, an amikacin nomogram builder based on PKPD criteria, has been developed and is freely available to help evaluating dosing recommendations.

## 1. Introduction

Amikacin is one of the most effective aminoglycoside antibiotics used against severe gram-negative bacterial infections and initial empirical antimicrobial treatments. It is commonly administered with β-lactam antibiotics to extend the antimicrobial activity spectrum against multidrug-resistant pathogens such as *Pseudomonas aeruginosa* [1,2]. Optimizing amikacin treatment has been recently proposed by clinicians, driven by the increase of resistance to alternative antibiotic drugs [3].

Amikacin treatment success, in terms of bacterial killing and clinical response, has been mainly associated with reaching a maximum concentration/minimum inhibitory concentration (Cmax/MIC) ≥ 8–10 with recommended Cmax target values of 40–64 mg/L [4,5,6,7,8,9,10]. A ratio between the area under the concentration-time curve (AUC) and the MIC (AUC/MIC) ≥ 70 (up to 80–100 in amikacin monotherapy or in critically ill patients with high-bacterial burden infections) and a time for which concentration exceeds the MIC (T_>MIC_) of at least 60% of the dosing interval administration has also been proposed as predictive efficacy pharmacokinetic/pharmacodynamic (PKPD) criteria [4,6,11]. AUC/MIC criteria has been more commonly applied for vancomycin and fluoroquinolones while an insufficient T_>MIC_ has been associated with an increase in the likelihood of resistance development and treatment failure of β-lactams or amikacin [4,11,12]. On the other hand, nephrotoxicity and ototoxicity continue to be a major concern associated with the clinical use of amikacin. A target value of minimum concentration (Cmin) lower than 4 mg/L has been proposed to reduce the risk of developing these toxicities [11,13].

Amikacin is characterized by a narrow therapeutic index and a large intra and interindividual pharmacokinetic (PK) variability associated with renal function, bodyweight, albumin or age, among others (Appendix A) [14,15,16,17,18,19,20,21,22,23,24,25,26,27,28]. In particular, low plasma concentrations of albumin (ALB) have been proposed as a predictor of nephrotoxicity in hospitalized patients treated with intravenous amikacin [29]. However, the impact of serum albumin (ALB) concentrations on amikacin exposure, efficacy and safety has not been extensively studied and is still unclear.

Amikacin was approved at 15 mg/kg/day as a once-daily dose or divided in 2 or 3 equal doses administered at equivalent intervals for patients with normal renal function [30]. Since 1990s, the once-daily or extended interval dose regimen has been globally adopted, improving microbiological and clinical outcomes without greater incidence of associated toxicities [31,32]. Amikacin has a concentration-related bacterial killing increasing post antibiotic effect (PAE) up to 8 h and decreasing adaptive bacteria resistance which represent clinical advantages of a once-daily dosing regimen [1,31]. Although several studies have shown that amikacin extended interval dose regimen was as effective as multiple-dose per day regimens, with lower risk of toxicity, the once-daily dosing regimen should be used with caution in specific populations [13,31,33].

The “one-dose-fits all” treatment strategy is a common practice in antibiotic drugs. Amikacin 15–20 mg/kg/day once-daily has been lately adopted, only eventually adjusted based on renal function or age [7,8,9,10,34]. Furthermore, no increase in toxicity for amikacin administered at higher doses (25–30 mg/kg/day) than standard ones (15 mg/kg/day) in specific populations (severe sepsis, critically ill patients) has been shown [6,35,36]. Recently published international guidelines for amikacin dosing recommended doses up to 30 mg/kg/day [8,34]. Therefore, there is no consensus regarding the optimal dose of amikacin treatment. However, several studies have pointed out that individualized amikacin dosing strategies could improve clinical outcomes with no additional toxicity [37,38].

The aims of the present study were (i) to evaluate the potential efficacy and safety of amikacin dosage recommended by the current international guidelines, (ii) to create an interactive amikacin dosage nomogram tool based on PKPD criteria and (iii) to evaluate the impact of intrinsic factors on amikacin dosing regimens with special focus on ALB.

## 2. Materials and Methods

### 2.1. Pharmacokinetic Analysis

An amikacin population PK (PopPK) model previously developed was used. Amikacin PK was described by a one-compartment model with first order linear elimination, parameterized in terms of clearance (CL) and volume of distribution (V) (Equations (1) and (2), respectively). Total bodyweight (TBW), ALB, estimated glomerular filtration rate (eGFR) calculated with Chronic Kidney Disease Epidemiology (CKD-EPI) equation and co-medication with vancomycin showed a significant impact on amikacin PK. Interindividual variabilities (IIV) of CL and V were 28.3% and 10.4%, respectively, following a log-normal distribution (exponential model). Additional details of the PopPK model are provided in Pérez-Blanco JS et al. [14]
(1)$$\mathrm{CL}\;\left(L/h\right)=\left(0.525+4.78\;\times \;\mathrm{CKD}-\mathrm{EPI}/98\right)\;\times \;0.77^{\mathrm{VANCO}}$$
(2)$$V\;\left(L\right)=26.3\;\times \;{\left(\mathrm{ALB}/2.9\right)}^{-0.51}\;\times \left[1+0.006\;\times \;\left(\mathrm{TBW}-70\right)\right]$$
where CL (L/h) is the total amikacin clearance, CKD-EPI (mL/min) is the estimated glomerular filtration rate by CKD-EPI equation, VANCO represents co-medication with vancomycin (0 for absence of vancomycin and 1 for co-medication with vancomycin), V (L) is the amikacin volume of distribution, ALB (g/dL) is the serum albumin concentration, and TBW (kg) is the total bodyweight.

The maximum (Cmax)and minimum (Cmin) concentrations at steady state and the T_>MIC_ expressed as percentage of the dosing interval are shown in equations 3–5, respectively.
(3)$$C_{\max}^{ss}\;\left(\mathrm{mg}/L\right)=\;\frac{D/T\;\times \;\left(1-{\;e}^{-\;\mathrm{CL}/\mathrm{Vd}\times \;T}\right)}{\mathrm{Vd}\;\times \;\mathrm{CL}/\mathrm{Vd}\;\times \;\left(1-{\;e}^{-\;\mathrm{CL}/\mathrm{Vd}\;\times \;\tau }\right)}$$
(4)$$C_{\min}^{ss}\;\left(\mathrm{mg}/L\right)={\;C}_{\max}^{\mathrm{ss}}\;\times {\;e}^{-\;\mathrm{CL}/\mathrm{Vd}\;\times \;\left(\tau -T\right)}$$
(5)$$T_{>\mathrm{MIC}}\;\left(\%\right)=(T-\ln\;\left(\mathrm{MIC}/C_{\max}^{\mathrm{ss}}\right)/\;\left(\mathrm{CL}/\mathrm{Vd}\times \;\tau \right))\times 100$$
where D (mg) is the total dose of amikacin administered, T (h) is the duration of infusion, CL (L/h) is the clearance, Vd (L) is the volume of distribution, τ (h) is the dosing interval, MIC (mg/L) is the minimum inhibitory concentration, and T_>MIC_ is the time with concentrations exceeding the MIC (percentage of τ).

### 2.2. Amikacin Dosage Guidelines Evaluation

Amikacin dosage recommendation guidelines selected in this work were (1) European Committee on Antimicrobial Susceptibility Testing guidance 2020 (EUCAST), (2) Antimicrobial Therapeutic Guide Mensa 2020 (Mensa), (3) Aminoglycoside Dosing in Adults Guideline of State of Queensland 2018 (Queensland), (4) The Sanford Guide to Antimicrobial Therapy 2019 (Sanford) and (5) UpToDate^®^ 2020 electronic clinical resource (UpToDate).

Monte Carlo simulations were carried out in R software with specific computing packages (stats, plyr, dplyr and ggplot2) using the PopPK model previously described to evaluate the efficacy (Cmax/MIC and T_>MIC_) and safety (Cmin) of each dosage proposed in the selected guidelines [39]. Mean dose of amikacin and mean CKD-EPI of the respective ranges proposed by the guidelines were selected for simulation purposes (120 mL/min for the >80 mL/min classification). For the UpToDate guideline where a range of dosing interval was proposed (eGFR ≤ 30 mL/min), dosing intervals of 48 h and 36 h were selected for renal function values of 10–20 mL/min and 20–30 mL/min, respectively. ALB and TBW were evaluated in the ranges of 1–6 g/dL and 40–150 kg, respectively. Dose recommended was administered in one-hour infusion. Cmax/MIC ≥ 10 and T_>MIC_ ≥ 60% were selected as efficacy PKPD thresholds with a MIC of 4 mg/L. PTA was calculated for 1000 virtual subjects of 70 kg in absence of vancomycin administration and taking into account the interindividual PK variability (residual unknown variability was not considered in the simulations). Treatment response was defined as effective when the PTA ≥ 90% for efficacy criteria (Cmax/MIC and/or T_>MIC_) and as toxic when Cmin ≥ 4 mg/L at steady state.

### 2.3. AMKnom: Interactive Amikacin Nomograms

An interactive R-based application, AMKnom, was developed to perform interactive amikacin nomograms based on PKPD criteria and subject characteristics. AMKnom was developed in R through Shiny package for implementing interactive functions into the simulation environment [40]. AMKnom was divided in two main sections:A.Input menu (left panel): information was displayed in three tabs: (i) Patient: TBW (kg), ALB (g/dL), CKD-EPI (mL/min) and co-medication with vancomycin (yes/no); (ii) Treatment (time of infusion (h), dosing interval (h) and MIC (mg/L)) and PKPD thresholds (Cmax/MIC, T_>MIC_ and Cmin); (iii) Graphical settings. PK parameters and PKPD target values for the specific scenario defined (patient, treatment and PKPD thresholds) are summarized at the bottom of patient tab.B.Graphical output (main panel): amikacin dose expressed in mg/kg (black solid lines) required to reach the selected Cmax/MIC threshold at steady state (Treatment & PKPD tab of input menu) was calculated for all possible values of two of the following variables: TBW, ALB and CKD-EPI. For each combination of two variables, the remaining variable was fixed to the value introduced in the patient tab of the input menu. Three different tabs are defined based on the fixed variable (CKD-EPI, TBW and ALB). For each dosing scenario using the Cmax/MIC criterion to define the dose, the T_>MIC_ and Cmin were calculated. A green area was drawn when T_>MIC_ complied with the threshold defined in Treatment & PKPD tab (input menu). A red area was drawn when Cmin was equal or higher than the toxicity threshold (Cmin) defined in Treatment & PKPD tab (input menu). A specific scenario defined in input menu is represented across the three graphical representations.

The ranges of continuous covariates of the AMKnom application were defined in agreement with the data supporting the PopPK model implemented [14]: TBW: 40–150 kg, ALB: 1–6 g/dL and CKD-EPI: 30–200 mL/min. The typical subject was defined as follows: TBW of 70 kg, ALB of 4.0 g/dL, CKD-EPI of 120 mL/min and absence of vancomycin co-administration.

AMKnom has been designed to calculate the amikacin dose (mg/kg) for an extended interval dosing selected by the practitioner/researcher (24 h by default) to reach a Cmax/MIC ≥ 10 (modifiable) at steady state for each single combination of two of the following patients’ characteristics: ALB, TBW and CKD-EPI. In addition, for the dosages selected, T_>MIC_ and Cmin at steady state are calculated and colored in the region were the threshold is reached for each criterion (efficacy: T_>MIC_ ≥ 60%; toxicity: Cmin ≥ 4 mg/L).

### 2.4. Evaluation of the Impact of Intrinsic Factors on Amikacin Dosing Regimens

AMKnom outputs are also helpful for better understanding of the quantitative impact of physiological factors on amikacin exposure and probability of treatment success. Deterministic simulations were carried out using AMKnom assuming absence of vancomycin administration and one of the following characteristics: TBW of 70 kg, ALB of 4.0 g/dL CKD-EPI of 120 mL/min; this was to evaluate the change in amikacin dosing requirements along the two variables not fixed in order to achieve the PKPD criteria of Cmax/MIC ≥ 10, T_>MIC_ ≥ 60% and Cmin < 4 mg/L at steady state.

## 3. Results

### 3.1. Amikacin Dosage Guidelines Evaluation

Large discrepancies were observed in amikacin dosage recommendations across the five dosing guidelines evaluated suggesting additional considerations might be helpful for optimizing amikacin dosing regimens (Table 1).

Treatment efficacy (Cmax/MIC ≥ 10 and T_>MIC_ ≥ 60%) and toxicity (Cmin ≥ 4 mg/L) versus ALB for the different dosing recommendations of each guideline stratified by renal function stages are shown in Figure 1. Overall, EUCAST and Queensland guidelines showed an adequate amikacin treatment efficacy (PTA > 90% for Cmax/MIC criterion) at the dosages recommended, independently of renal function stage and considering normal ALB. On the other hand, EUCAST and Queensland dosing recommendations were also associated with a high probability of expected toxicity in renal dysfunction subjects (PTA toxicity > 50% for any ALB and eGFR < 40 mL/min). Sanford and UpToDate guidelines showed acceptable safety profiles across the different dosage scenarios evaluated (PTA toxicity < 50%). However, potential efficacy issues might be expected due to the low doses or inadequate dosing interval recommended by these guidelines. Furthermore, at the dosage recommended, mainly based on renal function status, the higher is the ALB, (i) the higher will be the PTA for Cmax/MIC criterion, (ii) the lower will be the PTA for T_>MIC_ efficacy criterion and (iii) the lower will be the PTA for toxicity. For all evaluated dosing regimens, patients with hypoalbuminemia (ALB < 3.5 g/dL) were at greater risk of toxicity, especially when combined with impaired renal function.

### 3.2. AMKnom: Interactive Amikacin Nomograms

A user-friendly web application, AMKnom, has been developed and is freely available at http://shiny.cumulo.usal.es/amknom/ (accessed on 4 February 2021). AMKnom allows an easy evaluation of amikacin dosage based on patient and treatment information together with efficacy and safety PKPD criteria. An amikacin nomogram can be computed and downloaded from the application for the specific scenario defined. Dynamic simulations can be performed by activating the “play button” for each covariate included in the simulations. This action allows to observe changes in amikacin dosing regimen over the range of a determined variable together with the impact on treatment efficacy (T_>MIC_ ≥ 60%) and toxicity (Cmin ≥ 4 mg/L). An example of the AMKnom application is shown in Figure 2. Considering the interactive properties of the AMKnom application, we recommend exploring the online tool to better understand the features implemented and the potential of the tool developed on improving amikacin dosage optimization based on PKPD criteria.

Influence of TBW, ALB and CKD-EPI on amikacin dose (mg/kg) once-daily administered in absence of vancomycin co-administration for a typical subject (TBW: 70 kg, ALB: 4 g/dL, CKD-EPI: 120 mL/min) is shown in Figure 3. These results show a significant influence of ALB on amikacin dosing optimization with a larger impact on treatment efficacy than CKD-EPI, the latest being currently the main driver of amikacin dosage selection. For instance, to reach a Cmax/MIC ≥ 10 in a typical subject, a 36% increase in dose must be considered for a change in ALB from 4 g/dL to 2 g/dL while an 8% dose reduction would be needed if the renal function decreases from 120 mL/min to 60 mL/min. Moreover, a decrease of 1 mg/kg/day (8% dose reduction) when TBW increases by 10 kg, from 70 kg to 80 kg, in a typical subject, would be necessary to achieve the defined efficacy Cmax/MIC threshold. This change in dosing selection based on TBW modifications had a higher impact in subjects with low TBW (15% dose decrease from 40 kg to 50 kg) than in heavier subjects (3% dose decrease from 140 kg to 150 kg).

## 4. Discussion

Despite continued progress toward “one-dose-fits-all” in antibiotic treatment strategies, amikacin dosage regimens should be further individualized based on efficacy and safety PKPD criteria together with individual patient information and clinical evolution. In the last few years, model-informed precision dosing (MIPD) has been shown as a promising tool to increase treatment success [41]. This methodology integrates PK information together with patient characteristics and exposure-response or PKPD relationships in an easy-to-use model-informed framework.

Antimicrobial therapy recommendations based on total and adjusted body weight, renal function or age present a wide variability across the international guidelines evaluated in this work (Table 1). Our results showed that intrinsic factors such as TBW, ALB and CKD-EPI had a significant impact on efficacy and safety of amikacin treatments. Considering these factors, amikacin 15–20 mg/kg/day once-daily regimen, lately adopted for most of antibiotics treatment, would reach efficacy target with limited probability of toxicity for a large majority of patients. However, this dosage could potentially increase the probability of toxicity in severe renal dysfunction stages, even more pronounced when associated with low ALB levels, and could also increase the risk of lack of efficacy in patients with hypoalbuminemia.

Renal elimination plays a key role in the PK of water-soluble drugs such as amikacin, primarily eliminated by the kidneys [42]. The majority of the antimicrobial therapy guidelines evaluated in this work recommended to adjust amikacin dose and frequency of administration based on renal function calculated with Cockcroft–Gault, CKD-EPI and Salazar–Corcoran equations [7,8,9,10]. However, recent studies have proposed CKD-EPI and revised Lund–Malmö as the best equations to characterize amikacin elimination in most of the physiopathological situations [43,44]. On the other hand, Queensland and EUCAST guidelines recommended the same initial single dose (mg/kg) for critically ill/febrile neutropenic patients with or without chronic renal impairment followed by TDM adjustment. Our results showed that a reduced amikacin dose of 8% was required when administered once-daily to reach treatment success in terms of efficacy comparing subjects with normal renal function (eGFR = 120 mL/min) versus mild renal impairment subjects (eGFR = 60 mL/min). These results were aligned with EUCAST and Queensland guidelines recommending reduced amikacin dose adjustment for a wide range of renal function. In general, across the renal function stages evaluated, EUCAST and Queensland showed an expected adequate treatment efficacy based on Cmax/MIC criterion requiring higher doses for eGFR > 50 mL/min to reach a T_>MIC_ ≥ 60%. It is important to highlight that special populations such as pediatrics, obese or renal failure patients (eGFR < 15 mL/min) were not considered in this work.

As other hydrophilic antimicrobials, amikacin exhibits a volume of distribution limited to the extracellular space, which is significantly affected by decrease in ALB concentrations [45]. Increase in the volume of distribution is likely to decrease Cmax and total amikacin concentrations over time. In consequence, it is important to considerate ALB for optimizing amikacin dosage in order to achieve the efficacy and safety PKPD targets proposed, avoiding potential sub-therapeutic concentrations in patients with hypoalbuminemia. Several disease stages are associated with hypoalbuminemia, such as analbuminemia, starvation, liver or renal disease, cancer, stress response, burns, trauma, surgery or septic shock [21,23,46,47,48,49,50]. Although some of the antimicrobial therapy guidelines recommend special caution in situations with increased volume of distribution such as sepsis or septic shock, cystic fibrosis, severe burns or ascites, there are no special considerations based on ALB concentrations [7,8,10]. Our findings show a significant influence of ALB on amikacin dosage regimens: considering the same renal function, patients with hypoalbuminemia should receive higher doses than normoalbuminemic ones for warrantying a comparable treatment efficacy. Increasing dose in patients with hypoalbuminemia may involve toxicity issues, as previously suggested, which could be potentially improved by extending the dosing interval (≥36 h) [29].

Amikacin dosing has been based on weight (mg/kg) for decades. TBW, ideal bodyweight (IBW), lean bodyweight (LBW) and adjusted bodyweight (ABW) have been proposed regardless body mass index (BMI) classification for dose calculations by many authors without consensus (most of them for critically ill and obese patients) [2,6,15,36,37,51,52,53,54,55]. Similarly, antimicrobial guidelines also have no consensus on the weight measure best describing the variability of amikacin PK. Exceptions to the use of TBW for amikacin dosing are described in the guidelines evaluated (i.e., ABW above 30 mg/kg/m^2^) in addition to new body size scalars proposed, such as skeletal muscle area, as better predictor of interpatient PK variability [56]. Amikacin dosing recommendations based on TBW seem to be valid for normal-weight patients (BMI < 25 kg/m^2^) (Appendix A) [9,57]. However, in overweight and obese patients (BMI ≥ 25 kg/m^2^) or physiopathological stages such as sarcopenic patients with cancer, drugs’ distribution can be largely modified [58,59]. In the amikacin PopPK model used in this work, TBW was the best body measurement metric to describe amikacin PK, and therefore, TBW has been used for dosage evaluation and simulations [14]. Thus, results obtained in obese patients and its impact on dosage recommendation together with efficacy and toxicity associated with alternative body sizes proposed in each guideline must be considered carefully.

Amikacin is often used in combination with other drugs, potentially modifying its PK and/or clinical outcome such as vasoactive drugs or nephrotoxic agents (vancomycin, furosemide or amphotericin B). Vasoactive drugs are commonly required to improve hemodynamic function in patients with sepsis or septic shock and can modify the extracellular fluid compartment and volume of distribution of water soluble antibiotics such as aminoglycosides [60]. In addition, total parenteral nutrition has been correlated with an expanded V and lower Cmax of amikacin as well as with an increase in creatinine clearance [61]. Apart from vancomycin, the effects of other drugs co-administered and parenteral nutrition were not taken into account in the present research. In consequence, additional considerations may be required in amikacin dose individualization when combined with additional supportive care therapies.

Aminoglycosides nephrotoxicity can limit their use in clinical practice. Amikacin presents nephrotoxicity, usually transient, and ototoxicity, commonly irreversible, related with high concentrations at the end of the dosing administration interval (Cmin > 4 mg/L) [62,63]. Dosage recommended by the amikacin guidelines evaluated in this work, except Sanford, presented toxicity issues (PTA > 30%) in moderate to severe decreased renal function patients (eGFR < 40 mL/min), the PTA for toxicity also becoming more pronounced for hypoalbuminemia subjects. Toxicity issues showed in patients with decreased CKD-EPI, low ALB or high TBW can eventually be managed by selecting and adequate dosing interval administration (≥36 h).

Monte Carlo simulations performed for the dosage recommended in the international amikacin guidelines evaluated in this work showed the high relevance of ALB for treatment efficacy and toxicity (Figure 1). Thus, a specific dosage can vary from absence to full achievement of efficacy and/or toxicity target only due to the ALB value within a patient. In addition, impact of changes in ALB, from 4 g/dL to 2 g/dL, comparing with changes in renal function stage, from 120 mL/min to 60 mL/min, showed a significant higher impact of ALB than renal function on dose adjustment required to reach treatment efficacy (36% increase vs. 8% reduction, respectively). These results highlighted the larger impact of changes in V accounted for ALB than the CL changes due to eGFR modifications on Cmax/MIC efficacy criterion. These findings reveal the significant role of ALB in amikacin dosage optimization based on PKPD criteria and support its potential consideration for future updates of the principal international amikacin dosing guidelines.

Antimicrobial therapy guidelines establish general amikacin dosing recommendations of great utility for different population groups, whereas amikacin dosing model-based tools such as TMDx, InsightR or AMKdose and nomograms could be more precise in certain situations [14,64,65]. TDMx is an extremely powerful dosing tool able to provide precise probabilistic and Bayesian dosing together with optimal sampling recommendations for several antibiotics (amikacin, vancomycin, tobramycin, etc.) [64]. In comparison with the probabilistic dosing module of TDMx, AMKnom reports an additional efficacy PKPD target (T_>MIC_), it is applicable in a wider population (PopPK model used to calculate individual PK exposures), and it allows to evaluate the impact of different variables on amikacin initial dosage recommendation due to the nomogram interface. Thirion et al. have recently published a nomogram of amikacin dosing in adult cystic fibrosis population based on Cmax/MIC criterion considering 20–45 mg/kg/day as efficacy dosages with a relevant impact of TBW and creatinine clearance [66]. Unfortunately, the impact of ALB on amikacin dosage was not evaluated. Thus, an interactive amikacin dosing evaluation and nomogram builder (AMKnom, http://shiny.cumulo.usal.es/amknom/ (accessed on 4 February 2021)) has been developed and is freely available. Accounting for patients’ characteristics together with efficacy and safety criteria, AMKnom allows the researchers and clinicians to evaluate different dosing regimens (i.e., q12h, q36h), PKPD thresholds (Cmax/MIC, T_>MIC_, Cmin) and MIC values. AMKnom implements an improved dosing selection algorithm principally driven by Cmax/MIC showing the results as a nomogram for easier application in the clinical routine in comparison with previous tools developed by our group such as AMKdose [14].

Despite the potential benefits and strengths of AMKnom in optimizing amikacin initial dosage based on PKPD criteria, some limitations due to the PK model implemented and the simulations performed might be acknowledged regarding its application in the clinical practice. Although the PopPK model implemented in AMKnom was successfully validated both internally and externally, the interactive nomogram has not been clinically validated. The magnitude and applicability of the amikacin initial dosage recommendation are directly impacted by the model used for individual PK exposure calculations, and consequently, model limitations and special considerations (doses and covariates ranges) might be carefully taken into account when applying precision dosing tools such as AMKnom (Appendix A) [14,21,22,25,67,68,69,70]. In addition, specific populations where significant PK differences have been shown, such as pediatric, overweight and obese (IMC ≥ 25 kg/m^2^), burns, amputees, patients with specific additional supportive care therapies (i.e., vasoactive drugs, parenteral nutrition) or end-stage renal disease (eGFR < 15 mL/min) were not considered in this work. Initial amikacin dosage recommended by AMKnom for patients suffering drastic physiological changes (ICU patients) should be considered carefully as time-varying behavior of covariates (eGFR, ALB, TBW) was not taken into account. Moreover, AMKnom has been designed for a priori amikacin initial dosage optimization and not for dosage adjustment. As long as individual PK information is available, different strategies such as TDM and Bayesian forecasting are recommended instead of AMKnom. Therefore, additional studies including the previously mentioned populations and clinical validation are seen as potential improvements of the interactive nomogram developed for amikacin initial dosage individualization.

## 5. Conclusions

The present study shows the efficacy and toxicity associated to the amikacin dosage recommended by the main international dosing guidelines together with the impact of total bodyweight, renal function and serum albumin on amikacin model-informed precision dosing recommendation. Efficacy and safety criteria of amikacin dosage recommended by the guidelines, based on renal function, were not achieved simultaneously in most of the clinical scenarios evaluated. In patients with hypoalbuminemia, the PTA of efficacy and safety were reduced due to an increased V, highlighting the influence of ALB on amikacin dosing optimization with a larger impact than eGFR. Thus, accounting for albumin effect might be of interest for future amikacin guidelines updates. AMKnom, a useful interactive amikacin nomogram builder based on PKPD criteria, mainly driven by Cmax/MIC and reporting T_>MIC_ as a complementary efficacy criterion and patient characteristics, including albumin, is freely available and can be helpful for initial amikacin dosing evaluation and dosage requirement in specific physiological status (i.e., hypoalbuminemia).

## Figures and Tables

**Figure 1 pharmaceutics-13-00264-f001:**
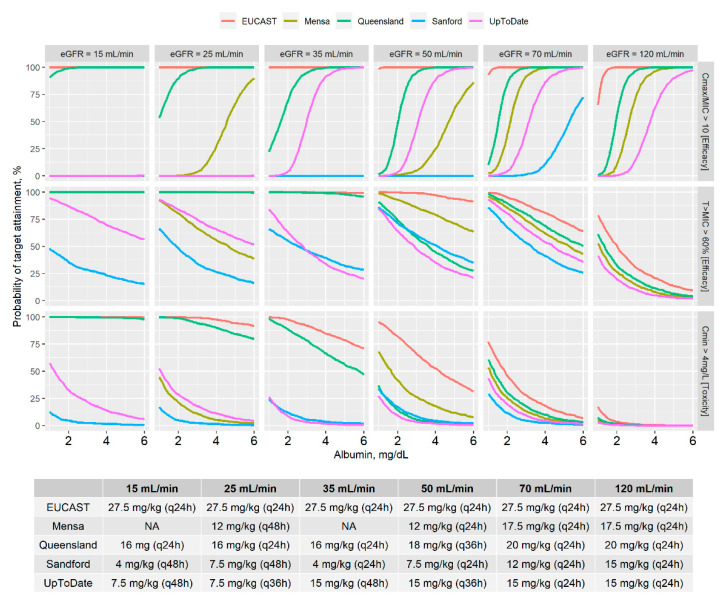
Efficacy and toxicity probability (upper panel, graphical representations) of amikacin dosage recommended (bottom table) by the Committee on Antimicrobial Susceptibility Testing guidance EUCAST, Mensa, Queensland, Sanford and UpToDate guidelines stratified by estimated glomerular filtration rate (eGFR). Cmax, maximum concentration; MIC, minimum inhibitory concentration; T > MIC, time with concentrations exceeding the MIC (percentage of dosing interval); Cmin, minimum concentration; 1000 virtual subjects simulated with total body weight of 70 kg, no vancomycin administration, infusion time of 1 h and amikacin administered once daily; MIC = 4 mg/L; NA, non-applicable; q24h, q36h and q48h are the dosing administration interval (24 h, 36 h and 48 h, respectively).

**Figure 2 pharmaceutics-13-00264-f002:**
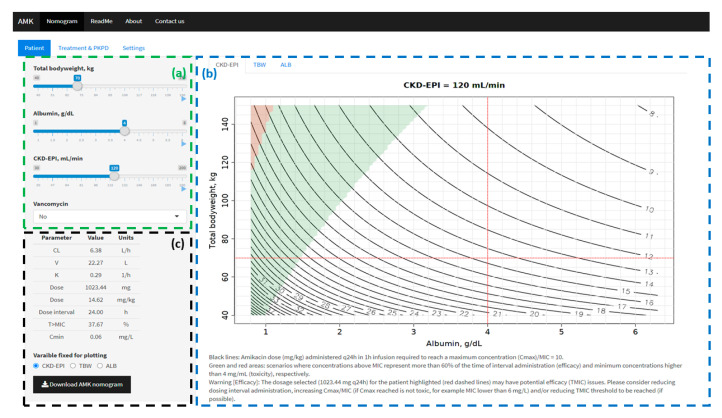
AMKnom application. (**a**) Input menu (3 tabs). (**b**) Graphical output: Title indicates the value of a specific variable fixed when simulations are performed; red dashed lines, subject defined in patient tab of input menu (70 kg, 4 g/dL of albumin and 120 mL/min); black solid lines, amikacin dose (mg/kg) required to achieve the Cmax/MIC threshold defined in Treatment & PKPD tab of (**a**) (Cmax/MIC = 10); green area, efficacy scenarios where time above MIC is equal or higher than the threshold defined in Treatment & PKPD tab of (**a**) (T_>MIC_ = 60%); red area, toxic scenarios where Cmin is equal or higher than the threshold defined in Treatment & PKPD tab of (**a**) (Cmin = 4 mg/L). (**c**) PK and PKPD results for the scenario defined in the input menu together with a download button of the three possible nomograms performed.

**Figure 3 pharmaceutics-13-00264-f003:**
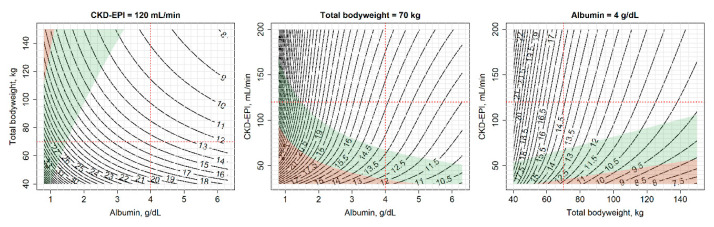
Amikacin dose (mg/kg) administered once-daily in 1 h infusion required to reach a maximum concentration (Cmax)/MIC = 10 across albumin, total body weight and estimated glomerular filtration rate (calculated with CKD-EPI equation) variations in the absence of vancomycin concomitant administration. Title of each plot, value of the variable fixed when simulations are carried out; red dashed lines, subject of 70 kg, 4 g/dL of albumin and 120 mL/min; MIC = 4 mg/L; green and red areas, scenarios where concentrations above MIC represent 60% of the time of dosing interval administration (efficacy) and minimum concentrations higher than 4 mg/mL (toxicity), respectively.

**Table 1 pharmaceutics-13-00264-t001:** Amikacin dosing recommended by the international dosing guidelines stratified by renal function.

Creatinine Clearance (mL/min)							
Guide	≤10	10–20	20–30	30–40	40–60	60–80	>80
EUCAST	25–30 (24)	25–30 (24)	25–30 (24)	25–30 (24)	25–30 (24)	25–30 (24)	25–30 (24)
Mensa	7.5–10 (48)	-	12 (48)	-	12 (24) ^ƍ^	15–20 (24)	15–20 (24)
Queensland	16 ^Δ^	16 ^Δ^	16 ^Δ^	16 ^Δ^	16–20 (36)	20 (24) ^♦^	20 (24) ^♦^
Sanford	3 (72) ^†^	4 (48)	7.5 (48)	4 (24)	7.5 (24)	12 (24)	15 (24)
UpToDate	7.5 (48–72) ^¶^	7.5 (24–72) ^¶^	7.5 (24–72) ^¶^	15 (48)	15 (36)	15 (24)	15 (24) ^&^

Amikacin doses are expressed in mg/kg and interval dosing in hours between brackets. EUCAST: Initial doses based on ideal bodyweight in seriously ill patients prior to therapeutic drug monitoring and dose adjustment. Renal function calculated by CKD-EPI equation [34]. Mensa: Amikacin dose based on adjusted bodyweight. Renal function calculated by Cockcroft–Gault equation [7]. Queensland: Amikacin dose based on ideal bodyweight or total bodyweight whichever is lower. Use ideal bodyweight for overweight patients (BMI ≥ 25 & < 30 kg/m^2^) and adjusted bodyweight for obese patients (BMI ≥ 30 kg/m^2^). Renal function calculated by an easily available estimated glomerular filtration rate equation. Cockcroft–Gault equation is not recommended [8]. Sanford: Amikacin dose based on ideal bodyweight for non-obese patients (BMI < 30 kg/m^2^). Use adjusted bodyweight in obese patients (BMI ≥ 30 kg/m^2^). Renal function calculated by Cockcroft–Gault equation using ideal bodyweight for non-obese patients and Salazar-Corcoran equation for obese patients [9]. UpToDate: Amikacin dose based on total bodyweight for underweight patients (BMI < 18.5 kg/m^2^); ideal bodyweight for patients with total bodyweight 1–1.25 folds’ ideal bodyweight and adjusted bodyweight for patients with total bodyweight > 1.25 folds’ ideal bodyweight. Renal function calculated by Cockroft–Gault equation using lean bodyweight [10]. ƍ Amikacin dosage only for creatinine clearance 40–50 mL/min. Δ Single dose only. Further doses should be under therapeutic drug monitoring (TDM). ♦ For critically ill and febrile neutropenic patients, administer a single dose of 30 mg/kg and monitor concentrations. † Amikacin postdialysis dose should be administered. ¶ Amikacin dosage based on serum concentrations. & Use traditional intermittent dosing for creatinine clearance > 120 mL/min.

## Data Availability

No new data were created or analyzed in this study. Data sharing is not applicable to this article.

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
