# Peer review of "Evaluation of Current Amikacin Dosing Recommendations and Development of an Interactive Nomogram: The Role of Albumin"

_pharmaceutics, 2021, doi:10.3390/pharmaceutics13020264_

Round 1

Reviewer 1 Report

The authors report on the development of AMKnom, an interactive amikacin nomogram based on their recently published popPK model (Pérez-Blanco et al. 2020 J Antimicrob Chemother). The nomogram was used to quantify the impact of covariates on amikacin exposure and the probability of treatment success for specific doses. The software tool could be used for a priori dose optimisation, but a posteriori dose optimisation (based on Bayesian forecasting) is not facilitated. Furthermore, a narrative review of five amikacin dosing guidelines was performed, illustrating large discrepancies. The study addresses a clinically relevant question, the design is straightforward, and the methods and results are clearly explained. Nevertheless, a critical interpretation of the work is lacking. Furthermore, a quantitative comparison with other popPK models would improve the relevance of the manuscript. Please consider the following comments and suggestions:

  1. How does the AMKnom tool differ from the previously published AMKdose tool, and how are conclusions in the current manuscript novel as compared to the results in the publication in JAC in 2020 (doi: 10.1093/jac/dkaa158)?
  2. The authors work with their own model, but other models are available as well. A quantitative comparison of the covariate effects and dose predictions between different models would be of great interest.
  3. How does AMKnom relate to the TDMx amikacin precision dosing tool (tdmx.eu), which is based on the model of Romano et al. and includes covariates creatinine clearance, trauma, weight, and sepsis.
  4. The authors focus on a priori dose optimisation based on covariates, including the eGFR. Are PK parameters (and thus exposure and target attainment) calculated based on the eGFR on the same day (after the dose was given, and thus not practical for dose optimisation) or on the previous day (and thus available for actual prediction)? How were time variations in covariates (mostly relevant for eGFR) handled in the model?
  5. While the covariates explain interindividual variability in CL and V, the remaining unexplained variability is significant (28.3% on CL and 10.4% on V). How was the resulting uncertainty in exposure and target attainment taken into account in the analyses? What about the residual variability?
  6. Please refer to the software tools correctly. R is the software, RStudio is the GUI for working with R. Which R-packages were used for the Monte-Carlo simulations?
  7. A limitations paragraph is missing in the manuscript Discussion. Please consider my previous comments for writing this limitations section. Also, the assumptions that are intrinsic to the model and their impact on the (reliability of the) simulation results need to be addressed. Furthermore, it should be stressed that an external validation of the software tool (in fact of the popPK model) would be very valuable.
  8. Introduction: “However, several studies have pointed out that individualized amikacin dosing strategies could improve clinical outcomes with no additional toxicity [14,34,35].” References 34 and 35 do not support the statement. Reference 14 is the original publication of the amikacin popPK model used in the current manuscript. Please provide adequate references.

Reviewer 2 Report

This paper proposed the importance of the individually change of dose of amikacin, not “one-dose-fits all” treatment, and the authors developed the new application named “AMKnom”, which is helpful for amikacin dosing evaluation.

The following are the reviewer’s comments.

1. The author’s one of the most important results would be the impact of albumin concentration on the amikacin efficacy and safety; because their effects were larger than those of renal function, which is used in the guideline.
If there are some patient’s example which showed the relationship between albumin concentration and side-effects by amikacin, the authors should add them at “Introduction” or “Discussion”. It will helpful the author’s proposal. e.g. A M Contreras et al., (1994) Low serum albumin and the increased risk of amikacin nephrotoxicity  PMID: 8079062
2. Are there some reports that how many people (or How often) cause side effects with the administration of the current guidelines? Are there any differences, such as BMI, albumin, and etc., between patients with side effects and those who do not have side effects? If there are some information, the authors need to add the description at the Introduction.
3. Line 121; Where did the author get the patient information?  The authors need to add about the origin of the patient information.

Round 2

Reviewer 1 Report

The authors have addressed my comments adequately. I have no further questions. Congratulations on the excellent manuscript!